# MR Angiography in Assessment of Collaterals in Patients with Acute Ischemic Stroke: A Comparative Analysis with Digital Subtraction Angiography

**DOI:** 10.3390/brainsci12091181

**Published:** 2022-09-02

**Authors:** Brian Tsui, May Nour, Iris Chen, Joe X. Qiao, Banafsheh Salehi, Bryan Yoo, Geoffrey P. Colby, Noriko Salamon, Pablo Villablanca, Reza Jahan, Gary Duckwiler, Jeffrey L. Saver, David S. Liebeskind, Kambiz Nael

**Affiliations:** 1Department of Radiological Sciences, David Geffen School of Medicine at UCLA, Los Angeles, CA 90095, USA; 2Department of Neurology, David Geffen School of Medicine at UCLA, Los Angeles, CA 90095, USA; 3Department of Neurosurgery, David Geffen School of Medicine at UCLA, Los Angeles, CA 90095, USA

**Keywords:** MRI, stroke, collaterals

## Abstract

Collateral status has prognostic and treatment implications in acute ischemic stroke (AIS) patients. Unlike CTA, grading collaterals on MRA is not well studied. We aimed to evaluate the accuracy of assessing collaterals on pretreatment MRA in AIS patients against DSA. AIS patients with anterior circulation proximal arterial occlusion with baseline MRA and subsequent endovascular treatment were included. MRA collaterals were evaluated by two neuroradiologists independently using the Tan and Maas scoring systems. DSA collaterals were evaluated by using the American Society of Interventional and Therapeutic Neuroradiology grading system and were used as the reference for comparative analysis against MRA. A total of 104 patients met the inclusion criteria (59 female, age (mean ± SD): 70.8 ± 18.1). The inter-rater agreement (*k*) for collateral scoring was 0.49, 95% CI 0.37–0.61 for the Tan score and 0.44, 95% CI 0.26–0.62 for the Maas score. Total number (%) of sufficient vs. insufficient collaterals based on DSA was 49 (47%) and 55 (53%) respectively. Using the Tan score, 45% of patients with sufficient collaterals and 64% with insufficient collaterals were correctly identified in comparison to DSA, resulting in a poor agreement (0.09, 95% CI 0.1–0.28). Using the Maas score, only 4% of patients with sufficient collaterals and 93% with insufficient collaterals were correctly identified against DSA, resulting in poor agreement (0.03, 95% CI 0.06–0.13). Pretreatment MRA in AIS patients has limited concordance with DSA when grading collaterals using the Tan and Maas scoring systems.

## 1. Introduction

In patients with acute ischemic stroke (AIS) due to a large vessel occlusion (LVO), robust collaterals have been identified as an important variable with prognostic and treatment implications [1,2,3]. AIS patients with good collaterals often present with smaller infarction core [4], slower infarct growth [5], improved recanalization rate, and eventually, better functional outcomes [6,7].

Collateral imaging has been evaluated by a variety of neuroimaging techniques. Vessel imaging can be performed by digital subtraction angiography (DSA), which is often considered the gold standard due to its high spatial and temporal resolution, or via non-invasive cross sectional modalities, such as CTA or MRA. In these techniques, collateral status is evaluated by assessing the number, size and rate of filling of the collateral vessels [8]. Cerebral perfusion techniques, such as CTP [9] or MRP [10] can also evaluate collaterals indirectly by assessing the efficiency of collateral perfusion. 

Non-invasive assessment of collaterals via CTA has been extensively evaluated with promising results [11,12,13]. In fact, several scoring systems for the determination of collaterals have been developed on CTA, including the Tan score [14], Maas score [15], and a regional leptomeningeal score [3]. Assessment of collaterals by MRA has been limited to a handful of studies [16,17,18,19] due to the limited use of pretreatment MRIs in stroke patients. However, with extension of the treatment window of acute ischemic stroke to 24 h and with the increased value of MRI in this group of patients [20,21], more pretreatment MRI/MRA will be performed. For institutions that utilize MRI routinely in pretreatment assessment of AIS patients, the clinical utility of MRA in collateral assessment requires further investigation.

In this study, we aimed to assess collaterals on pretreatment MRA in AIS patients with anterior circulation proximal arterial occlusion using two common scoring systems used in CTA (Tan score [14], Maas score [15]), and to perform a comparative analysis against DSA as the reference standard. 

## 2. Materials and Methods

### 2.1. Patients

This retrospective study was approved by our institutional review board. Consecutive patients with acute ischemic stroke were identified between 1 January 2010 and 31 August 2019 and included if they met the following inclusion criteria: (1) anterior circulation proximal arterial occlusion defined as occlusion or intracranial internal carotid artery (ICA) or middle cerebral artery (MCA) M1 or proximal M2 segments; (2) underwent pretreatment MRI including MRA; (3) underwent DSA and endovascular treatment. Patients were excluded if they had posterior circulation occlusion, ACA stroke, or poor image quality.

Clinical data including patient age, sex, last known well time, NIH stroke scale (NIHSS) score, hospital presentation time, and door-to-needle time were collected. In addition, treatment type, including intravenous tPA, endovascular therapy, degree of recanalization using thrombolysis in cerebral infarction (TICI) scores and modified Rankin score (mRS) at 90 days were noted when available. 

### 2.2. Image Acquisition:

For CE-MRA, a single echo 3D RF-spoiled gradient echo sequence was used to image the neck and head with the following parameters: TR/TE 3.34/1.31 ms; FA: 25°; FOV: 320 × 230 mm^2^; matrix: 576 × 478 mm^2^, 120 slices × 1 mm thick. Parallel imaging with an acceleration factor of 3 was applied. With these settings, a 3D volume with a voxel-size of 0.67 × 0.56 × 1 mm^3^ was obtained for a combined neck-brain coverage during a 26 s acquisition. Multislab TOF-MRA of the brain was performed with 5 axial slabs of 30 slices per slab, each 1 mm thick, with the following parameters: TR/TE: 25/3.86 ms; FA: 20°; matrix: 512 × 332 mm^2^; FOV: 210 × 184 mm^2^; and parallel imaging with an acceleration factor of 2 resulting in the acquisition of a 3D voxel-sizes of 0.41 × 0.55 × 1 mm^3^ during a 6-min 10-s acquisition. 

### 2.3. Image Analysis:

#### 2.3.1. MRA

Two neuroradiologists, with 6 and 8 years of post-fellowship experience, respectively, independently graded the MRAs based on the Tan score and Maas scores (Table 1). Both neuroradiologists were blinded to the clinical information and outcomes. 

Scores were then dichotomized to sufficient and insufficient collaterals as indicated in Table 1. Discrepancies between the two neuroradiologists were resolved by consensus, and the consensus scores were then used for comparative analysis against DSA as the reference standard. 

#### 2.3.2. DSA

A separate interventional neuroradiologist with 6 years of post-fellowship experience graded the collaterals using the ASITN/SIR collateral flow grading system [22] on baseline pretreatment DSA runs: grade 0: no collaterals; grade 1: slow collaterals to the periphery of the ischemic site; grade 2: rapid collaterals to the peripheral of the ischemic site with persistence of some of the defect; grade 3: collaterals with slow but complete angiographic blood flow to the ischemic bed by the later venous phase; grade 4: complete and rapid collateral blood flow to the entire ischemic territory. The interventional neuroradiologist was blinded to the clinical information. Patients were dichotomized as good collaterals (ASITN grades 3 and 4) and insufficient collaterals (ASITN grades 0, 1, and 2) for comparative analysis against MRA.

### 2.4. Statistical Methods

Baseline characteristics were compared between patients with insufficient and sufficient collaterals using univariate analysis. Interobserver agreement for MRA collateral score grading was assessed using a weighted kappa test with 95% CI. MRA collateral scores were compared against DSA scores using the chi-squared test. The accuracy statistics, including sensitivity, specificity, positive predictive value and negative predictive value, were reported. Significance level was set at *p* = 0.05. 

## 3. Results

Among the 150 patients evaluated, a total of 104 patients (59 female) were ultimately included (Figure 1). In 27 patients, the baseline (before thrombectomy) catheter angiograms were deemed insufficient for assessment of collaterals due to limited angiographic runs or inadequate head coverage. In an additional 13 patients, the pre-treatment MRA was nondiagnostic for the evaluation of collaterals.

The mean age was 70.8 ± 18.1 years (mean ± SD). There were 12 patients with occlusion in the internal carotid artery, 69 patients with occlusion in the M1 segment, and 23 patients with occlusion in the M2 segment. The time from symptom onset was 208.1 ± 198.4 min (mean ± SD). The severity of stroke as determined by the baseline NIHSS was 16 (9–20) (median, IQR). The time from imaging to groin puncture was 60 (40–84) minutes (median, IQR). There were 13 patients who were treated with tPA (12.5%). Based on DSA, there were 55 patients with insufficient collaterals and 49 patients had sufficient collaterals. The breakdown of clinical data is summarized in Table 2. 

There were 30 patients who had pre-treatment TOF-MRA and 74 patients who had pre-treatment CE-MRA. The interobserver agreement between the neuroradiologists in grading collaterals using the Tan score was *k* = 0.49 (95% CI 0.37–0.61), which was modestly improved after dichotomization, 0.52 (95% CI 0.36–0.67). The interobserver agreement in grading collaterals using the Maas score was *k* = 0.44 (95% CI 0.26–0.62), which was modestly improved after dichotomization, 0.46 (95% CI 0.18–0.73).

The agreement for defining collaterals between MRA and DSA was poor (0.09, 95% CI 0.1–0.28) using the Tan score. Using the Tan score, 35 out of the 55 patients (63%) with insufficient collaterals, and 22 out of 49 patients (45%) with sufficient collaterals were correctly identified on MRA. 

The agreement for defining collaterals between MRA and DSA was also poor (0.03, 95% CI 0.06–0.13) using the Maas score. Using the Maas score, 51 out of the 55 patients (93%) with insufficient collaterals and only 2 out of 49 patients (4%) with sufficient collaterals were correctly identified on MRA.

In the TOF-MRA subgroup (*n* = 30), 12 patients had sufficient collaterals while 18 patients had insufficient collaterals based on DSA results. The interobserver agreement in grading collaterals using TOF-MRA was *k* = 0.25 (95% CI 0.05–0.45) for the Tan score and *k* = 1 for the Maas score. Using consensus Tan scores, 17 out of 18 patients with insufficient collaterals were correctly identified, and 1 out of 12 patients with sufficient collaterals was correctly identified (*p* = 0.77). Using consensus Maas scores, 18 out of 18 patients with insufficient collaterals were correctly identified and 0 out of 12 patients with sufficient collaterals were correctly identified (*p* = 0.27). A summary of the sensitivity, specificity, positive predictive value, negative predictive value, and accuracy is provided in Table 3. An imaging example is shown in Figure 2.

In the CE-MRA subgroup (*n* = 74), 37 patients had sufficient collaterals while the other 37 patients had insufficient collaterals based on DSA results. The interobserver agreement in grading collaterals using CE-MRA was *k* = 0.50 (95% CI 0.31–0.68) using Tan scores and 0.43 (95% CI 0.16–0.71) for the Mass scores. Using consensus Tan scores, 18 out of 37 patients with insufficient collaterals were correctly identified and 21 out of 37 patients with sufficient collaterals were correctly identified (*p* = 0.64). Using consensus Maas scores, 33 out of 37 patients with insufficient collaterals were identified but only 2 out of 37 patients with good collaterals were correctly identified (*p* = 0.40). A summary of the sensitivity, specificity, positive predictive value, negative predictive value, and accuracy is provided in Table 3. An imaging example is shown in Figure 3.

In a subgroup analysis of patients who received IV tPA (*n* = 13, 12.5%), tPA administration was not identified as a contributing factor for the degree of agreement between DSA and MRA. Using the Maas score, 8 out of 9 with insufficient collaterals and 0 out of 4 with sufficient collaterals were correctly identified (*p* = 0.50). Using the Tan score, 6 out of 9 with insufficient collaterals and 0 out of 4 with good collaterals were correctly identified (*p* = 0.21). 

## 4. Discussion

Our results showed poor agreement and insufficient diagnostic performance of pretreatment MRA in determination of collaterals in comparison to DSA as the reference standard. IV tPA administration did not affect the results. We like to highlight a few important results below.

The Maas scoring system had very low sensitivity for detection of collaterals using MRA with 0% sensitivity for TOF-MRA and 5% sensitivity for CE-MRA. This is likely inherent to the design of this scoring system and the way dichotomized results are used. A patient will be scored as having sufficient collaterals only if the collaterals are equal to the contralateral side (score 3, 4, 5), resulting in very high specificity but at the cost of low sensitivity. In contrast, using the Tan scoring system, a patient with collaterals between 50–100% will be included in the sufficient group and hence higher sensitivity values. In our study, using Tan scores resulted in an increased sensitivity of 8% for TOF-MRA and 52% for CE-MRA in comparison to Maas scores.

Additionally, regardless of the scoring system used, TOF-MRA has a lower sensitivity in comparison to CE-MRA for detection of collaterals. Prior studies have shown low sensitivity of TOF in detection of collaterals due to insensitivity to slow and sluggish collaterals, a known limitation related to spin saturation and dephasing in TOF-MRA [23]. We showed an overall accuracy of 60% for TOF-MRA in detecting collaterals; however, this number is largely overstated due to a very high specificity to lack of collaterals on TOF-MRA. In the study by Boujan et al. [18], an accuracy of 21.5% was reported for TOF-MRA in determination of collaterals. The overall accuracy of CE-MRA in our study was also modest; 53% for Tan and 47% for Maas scores, lower than what was reported by Boujan et al. using the Tan score (88%) [18]. 

Finally the lower diagnostic performance of MRA, regardless of the scoring system applied or technique used (TOF vs. CE-MRA), can be explained by the lower spatial or temporal resolution of MRA data and lack of dynamic information. Approximating the diagnostic performance of DSA may be a tall task for MRA in its current form. In fact, prior reports on the use of single phase CTA in comparison to DSA have shown modest diagnostic accuracies ranging from 24–78% in a few studies [24,25], even though CTA in general has a higher spatial resolution compared to MRA. An additional study used the Calgary Collateral score with correlation to DSA, which demonstrated modest (rho = 0.43) correlation on single phase CTA [26]. Multiphasic CTA demonstrated improved accuracy (81–82%) [25,27]. 

Our imaging protocol in this study used a combined head and neck acquisition rather than a dedicated head MRA. The timing of image acquisition has significant effect on determination of collaterals, and late arterial imaging or early venous imaging may increase the sensitivity of detecting small leptomeningeal collaterals. As pial collaterals require time to fill, early arterial image acquisition can result in underestimation of the collaterals [3,28,29]. In our study, the timing of MRA acquisition was optimized for early arterial enhancement to evaluate both neck and proximal intracranial arteries. Despite a 26 s acquisition, we suspect that this relatively early arterial imaging may have had a negative effect in the assessment of small and distal leptomeningeal collaterals. One way to address this limitation and mitigate the effect of early arterial imaging is to perform multiphase MRA [16,30].

An additional reason for possible lower diagnostic performance of our MRA data is that the spatial resolution of our study (0.67 × 0.56 × 1 mm^3^ for CE-MRA and 0.41 × 0.55 × 1 mm^3^ for TOF-MRA) is lower than CTA, which may have limited the detection of small collaterals. Improving the spatial resolution can potentially improve this limitation; however, this needs to be balanced against acquisition time, an important factor in the pretreatment phase of stroke patients. Improved MRI hardware and pulse sequence design have the potential to increase spatial and temporal resolution while keeping the acquisition time constant [31]. 

Another important point to consider is that applying Maas or Tan scores that have been traditionally used on CTA may not sufficiently exploit the information that exists in MRA for detection of collaterals. It is plausible that using a more robust scoring scale dedicated to MRA, with the inclusion of signal intensity assessment [32] or flow data, can improve the diagnostic capability of MRA. Including information about parenchymal signal change over time during contrast circulation (“poor-man perfusion maps”) may be another alternative path that can provide added value to conventional MRA scores to overcome challenges associated with underestimating collateral status. Further work and validation to establish a MRA-based collateral scoring system may be needed. Potential avenues include previously described dynamic multiphase MRA and collateral maps [33]. In this proposed scoring system, patients with collateral maps showing a minimum MAC 3 score (less than half of the MCA territory in the capillary and early venous phases, or perfusion delay more than half of the MCA territory in the capillary phase but with minimal delay in early venous phase) demonstrated better outcomes compared to patients who had more severe perfusion delay. This information suggests that multiphasic MRA may be able to better prognosticate patients compared to single phase MRA.

## 5. Conclusions

In summary, in patients with AIS, pretreatment MRA has limited concordance with DSA when grading collaterals using the Tan and Maas scoring systems. Improvement in image acquisition, increasing resolution, applying multiphase imaging, and devising dedicated scoring systems for MRA may be needed to improve the diagnostic performance, particularly as the use of MRI becomes more common in the pretreatment phase of patients with AIS. 

## Figures and Tables

**Figure 1 brainsci-12-01181-f001:**
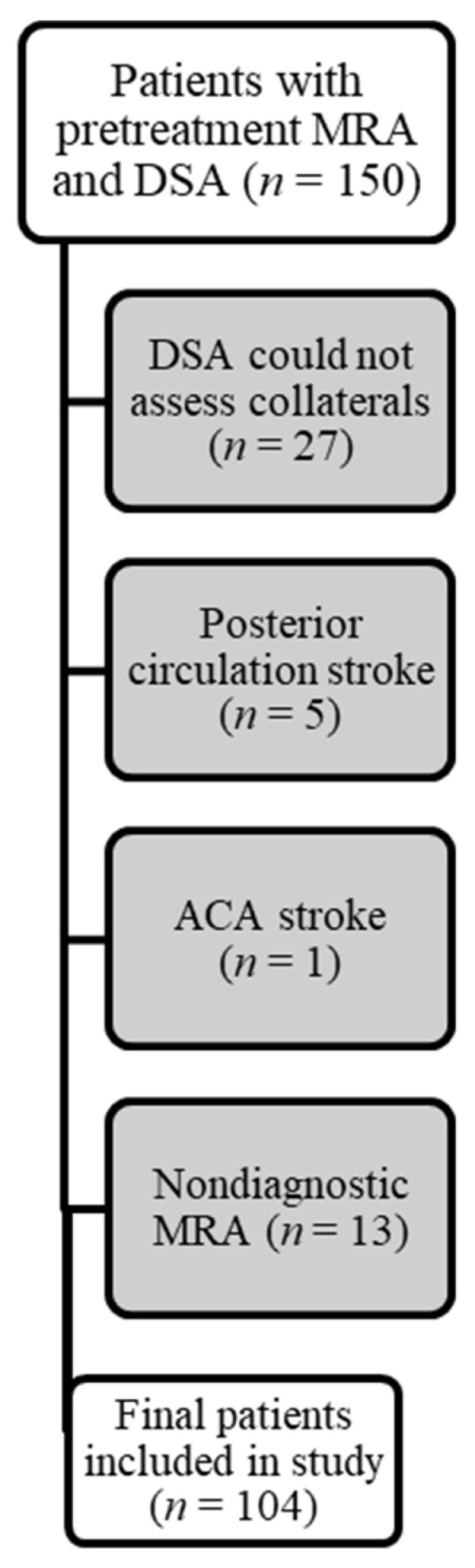
Flowchart of patient selection.

**Figure 2 brainsci-12-01181-f002:**
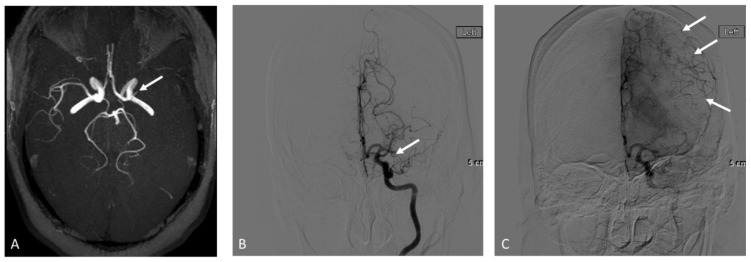
Adult patient who presented with right sided weakness and aphasia, NIHSS: 20. Axial maximum intensity projection from time-of-flight MRA shows left M1 occlusion (arrow in (**A**)) and paucity of collaterals along the left MCA territory. Early arterial phase from the follow up digital subtraction angiography confirms M1 occlusion (arrow in (**B**)) while later delayed phase imaging demonstrates good collaterals (arrows in (**C**)).

**Figure 3 brainsci-12-01181-f003:**
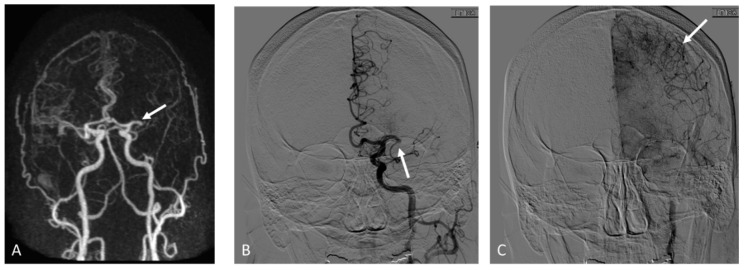
Adult patient who presented with right sided weakness and aphasia, NIHSS: 15. Coronal maximum intensity projection from contrast-enhanced MRA shows left M1 occlusion (arrow in (**A**)) and fewer than 50% collaterals along the left MCA territory in comparison to the right side. Early arterial phase from the follow up digital subtraction angiography confirms M1 occlusion (arrow in (**B**)) while later delayed phase imaging demonstrates robust collaterals (arrows in (**C**)).

**Table 1 brainsci-12-01181-t001:** Maas and Tan scoring definitions and dichotomization.

	Collaterals	Dichotomization
	**Maas scoring**	
1	Absent	Insufficient
2	Less than contralateral normal side	Insufficient
3	Equal to contralateral normal side	Sufficient
4	Greater than the contralateral normal side	Sufficient
5	Exuberant	Sufficient
	**Tan scoring**	
0	No collaterals (no filling of the occluded area)	Insufficient
1	Poor collaterals (>0% but <50% of the occluded area)	Insufficient
2	Moderate/good collaterals (>50% filling of the occluded area, <100%)	Sufficient
3	Equal (100% of the occluded area)	Sufficient

**Table 2 brainsci-12-01181-t002:** Demographic data of patients with poor and sufficient collaterals as defined by DSA.

	Poor Collaterals (*n* = 55)	Sufficient Collaterals (*n* = 49)	*p* Value
Age, mean (SD)	72.2 (18.9)	69.2 (17.1)	0.40
Sex (F/M)	34/21	25/24	0.31
LVO location (ICA/M1/M2)	8/35/12	4/34/11	0.80
NIHSS, median (IQR)	16 (10–20)	15 (9–19)	0.40
Time-from-stroke onset, mean (SD)	250.3/306.7	192.7/167	0.24
* 90-day mRS ≤ 2 (*n*)	16/49	12/39	0.85
Maas MRA score (correct)	51/55	2/49	0.48
Tan MRA score (correct)	35/55	22/49	0.38

* 90-day modified Rankin Score (mRS) was only available in 88 patients.

**Table 3 brainsci-12-01181-t003:** Sensitivity, specificity, PPV, NPV and accuracy of TOF-MRA and CE-MRA.

Modality	Score	TP (*n*)	TN (*n*)	FP (*n*)	FN (*n*)	Sensitivity	Specificity	PPV	NPV	Accuracy
TOF-MRA	Tan	1	17	1	11	0.08	0.94	0.50	0.61	0.60
	Maas	0	18	0	12	0	1	0	0.60	0.60
CE-MRA	Tan	21	18	19	16	0.57	0.49	0.53	0.53	0.53
	Maas	2	33	4	35	0.05	0.89	0.33	0.49	0.47

## Data Availability

The data presented in this study are available on request from the corresponding author. The data are not publicly available as the information is private health information from patients.

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
