# Peer review of "MR Angiography in Assessment of Collaterals in Patients with Acute Ischemic Stroke: A Comparative Analysis with Digital Subtraction Angiography"

_brainsci, 2022, doi:10.3390/brainsci12091181_

Round 1

Reviewer 1 Report

The topic is very important, the methodology of conducting the study is correct (albeight retrospective), but the the Authors' idea of ​​apply the monophasic CTA scores to MRA (contrast enhanced and TOF) as tool for leptmeningeal collateral evaluation and so for treatment decision of Patients with AIS is questionable. The fundamental limitations are  probably represented by the use of a score not dedicated to MR and the unavailability of newer sequences, such as 4D-MR angio (contrast enhanced or -even better- ASL based), as they clearly state in limitations section.

The lack of significant correlations between MRA and DSA is only marginal; probably the use of an original score to be validated against DSA would have added great value to the work.

The authors argue that MRI is part of the diagnostic workflow of patients who refer to the ED with AIS, but my personal consideration is that MR evaluation of collateral circulation is marginal aspect; most important informations are diagnostic confirmation of the suspected stroke and the exclusion of mimics, the correct evaluation of the site of occlusion, of the extension of the core and the mismatch (DWI / PWI), elements already sufficient to decide for treatment and formulate prognosis.

Reviewer 2 Report

The authors present data on MRA collateral scoring in comparison to gold standard DSA. They demonstrate that MRA collateral scoring in its present form is unsuitable and suggest and discuss topics for further research.

The main question addressed is whether collateral scoring by CEMRA using MRI is capable of providing sufficient information on collaterals in patients with AIS. 

The topic is relevant and interesting since it is expected that the volume of MRI employed in AIS patients increases since the time window for potential treatment has increased significantly and more patients become eligible.

The topic is relevant and interesting since it is expected that the volume of MRI employed in AIS patients increases since the time window for potential treatment has increased significantly and more patients become eligible.

The paper is very well written.  The text is easy to read and clear.

Reviewer 3 Report

Pretreatment 3D TOF and CE MRA was compared with DSA during subsequent EVT procedures for collateral status assessment in LVO AIS patients. The authors demonstrated clearly that (monophasic) MRA collateral scoring is not reliable  for correct rating of the collateral status of these patients (underestimation issue, low sensitivity). 

The manuscript is well written, with only few minor typo’s. The results with discussion and conclusion are presented in a clear way.

Although it is already common sense/published, that you should be very careful with interpretation of collaterals in single phase MRA (especially 3D TOF)/CTA, I think the manuscript is relevant though. Not only by adding clear scientific data, but also by contributing to the thinking/future research designs of how we can develop/tweak robust, day-to-day methods for optimal evt patient selection.

The potential reasons for poor collateral assessment with single phase MRA are discussed and some potential solutions are proposed in the discussion section. I only don’t fully agree with the lower spatial resolution issue. You don’t need info about occlusion sites, but only info about absence or presence of contrast inflow (or other markers for blod inflow) in areas at risk. E.g., this can be done by using color coded collateral maps based on multiphasic single slab MRA (ref 33, “poor man’s perfusion maps”). If needed, you can use fusion imaging tools for parenchymal or vascular correlation with other mri sequences.

The authors suggestions for improving future MRI protocols for collateral status scoring systems are a bit scanty. They might consider to improve this.  
